# Cilia, Centrosomes and Skeletal Muscle

**DOI:** 10.3390/ijms22179605

**Published:** 2021-09-04

**Authors:** Dominic C. H. Ng, Uda Y. Ho, Miranda D. Grounds

**Affiliations:** 1School of Biomedical Science, Faculty of Medicine, University of Queensland, St Lucia, Brisbane, QLD 4072, Australia; u.ho@uq.edu.au; 2School of Human Sciences, Faculty of Medicine, University of Western Australia, Perth, WA 6009, Australia

**Keywords:** myogenesis, primary cilia, proliferation, differentiation, satellite cells, cytoskeleton, extracellular matrix

## Abstract

Primary cilia are non-motile, cell cycle-associated organelles that can be found on most vertebrate cell types. Comprised of microtubule bundles organised into an axoneme and anchored by a mature centriole or basal body, primary cilia are dynamic signalling platforms that are intimately involved in cellular responses to their extracellular milieu. Defects in ciliogenesis or dysfunction in cilia signalling underlie a host of developmental disorders collectively referred to as ciliopathies, reinforcing important roles for cilia in human health. Whilst primary cilia have long been recognised to be present in striated muscle, their role in muscle is not well understood. However, recent studies indicate important contributions, particularly in skeletal muscle, that have to date remained underappreciated. Here, we explore recent revelations that the sensory and signalling functions of cilia on muscle progenitors regulate cell cycle progression, trigger differentiation and maintain a commitment to myogenesis. Cilia disassembly is initiated during myoblast fusion. However, the remnants of primary cilia persist in multi-nucleated myotubes, and we discuss their potential role in late-stage differentiation and myofiber formation. Reciprocal interactions between cilia and the extracellular matrix (ECM) microenvironment described for other tissues may also inform on parallel interactions in skeletal muscle. We also discuss emerging evidence that cilia on fibroblasts/fibro–adipogenic progenitors and myofibroblasts may influence cell fate in both a cell autonomous and non-autonomous manner with critical consequences for skeletal muscle ageing and repair in response to injury and disease. This review addresses the enigmatic but emerging role of primary cilia in satellite cells in myoblasts and myofibers during myogenesis, as well as the wider tissue microenvironment required for skeletal muscle formation and homeostasis.

## 1. Introduction

Cilia are evolutionarily conserved, microtubule-based cellular appendages that perform numerous functions required for normal development. Ciliary dysfunction underlies a broad spectrum of clinical conditions collectively known as ciliopathies. Cilia are composed of a core of microtubule bundles, termed axonemes, that are ensheathed in a ciliary membrane compartment and anchored at the base by the matured mother centriole or basal body (Figure 1A). In some specialised cells, such as spermatozoa and respiratory epithelia, the microtubule motor-driven biomechanical motion of cilia is required for fertility and to prevent chronic airway infections. Canonical axonomes in motile cilia are comprised of nine doublet microtubules arranged in a radial configuration surrounding two central microtubules in a 9 + 2 arrangement, whereas, in contrast, non-motile or primary cilia axonemes are typically 9 + 0 in structure and lack central singlet microtubules [1] (Figure 1B). These non-motile primary cilia are found in almost all other cell types, including muscle myoblasts and myofibroblasts, where their full spectrum of functions remains incompletely defined [2]. Originally believed to be vestigial organelles, it is now evident that primary cilia function as sensory appendages that co-ordinate cell signalling pathways to regulate proliferation, differentiation, cell polarity and cell–cell communication [3]. Indeed, a sensory role was proposed for cilia observed in avian muscle spindles that might detect fluid changes in periaxial spindle space and relate these changes to intrafusal myofibers [4]. In addition, primary cilia in various stem and progenitor populations contribute to tissue morphogenesis and homeostasis [5,6]. Skeletal muscle constitutes about 40% of an individual’s body mass and is fundamental to locomotion, temperature regulation and metabolism [7]. Thus, there is considerable interest in the molecular and cellular processes involved in forming skeletal muscle (myogenesis) during development and in situations of severe injury that required regeneration and new muscle formation, plus in context of neuromuscular diseases. This review focuses on the role of non-motile primary cilia (herein referred to simply as cilia) in skeletal myogenesis and discusses the recent revelations that ciliary appendages are intimately involved in the control of cell fate decisions, cell cycle timing and proliferative capacity in muscle.

## 2. Formation and Resorption of Cilia during Skeletal Myogenesis

Skeletal muscle is comprised of striated, multinucleated myofibers organised into compact bundles. During development, myogenesis is initiated when multi-potent mesodermal cells are committed to skeletal myoblasts. These myoblasts are muscle precursors that express Pax3 and Pax7 and exhibit a high proliferative capacity. Following a proliferative phase, the expression of the muscle regulatory family (MRF) of transcription factor proteins (MyoD, Myf5 and myogenin) coincides with cell cycle arrest, cessation of myoblast proliferation and initiation of differentiation, accompanied by alignment of the mononucleated myoblasts and their fusion to form multinucleated syncytia called myotubes that assemble contractile cytoskeletal units, known as sarcomeres (Figure 1C). These mature into myofibers that become innervated and are organised into bundles and are responsible for contractile force generation to enable movement.

In postnatal and adult skeletal muscle, a proportion of myoblasts expressing Pax7 (Pax7^+ve^), also referred to as satellite cells, are retained in a quiescent state and are located outside the myofiber lying between the sarcolemma and the external lamina (widely referred to as basal lamina); these cells initiate proliferation and generate myoblasts that are principally responsible for the regenerative capacity of skeletal muscle. When activated, some satellite cells undergo oriented asymmetric divisions to give rise to muscle-committed progenitors (myoblasts) and a self-renewed ‘stem cell’ population that remains in the ‘satellite’ position on the myofiber surface (Figure 1C) [8,9]. It is proposed that myoblasts positioned away from the myofiber surface undergo further rounds of amplifying division before exiting the cell cycle, undergoing differentiation and cell fusion to form myotubes that can fuse with existing myofibers or form new myofibers for muscle repair (in response to severe muscle damage that results in myonecrosis). It is increasingly appreciated that a reduced proliferative capacity of satellite cells and abnormalities in these myogenic processes can contribute to muscle ageing and disease progression. In skeletal muscle tissue, cilia are found on mononucleated cells that include satellite cells/myoblasts and resident fibro/adipogenic progenitors in the interstitial connective tissue [10,11]. The presence of cilia on myoblasts was described 50 years ago in ultrastructural studies for avian and then human myoblasts in culture [12,13] and in regenerating adult mouse muscle [14]. More recent studies utilising fluorescence imaging and the tracking of ciliary membrane or axonemal components in primary cultures of mouse myoblast or ex vivo cultures of myofibers have revealed more precise dynamics of cilia formation (ciliogenesis) during myogenesis [10,15].

In primary myoblast cultures, following serum withdrawal, ciliogenesis coincides with cell cycle exit and the early onset of differentiation (Figure 1C) [15]. This is consistent with the notion that the formation of cilia is an anathema to cell division. For example, cilia disassembly is required prior to mitosis, as the mother centriole that comprises the basal body of cilia is required for spindle assembly [16]. Moreover, mitotic kinases required for spindle assembly, such as Aurora A, conversely promote the disassembly of cilia [17]. With the progression of differentiation and prior to myoblast fusion, cilia on myoblasts are disassembled and are largely absent in mature, multinucleated myofibers (Figure 1C) [15]. Coincidentally, it is also during this time that myoblasts undertake a process of centrosome reduction to facilitate a switch to non-centrosomal microtubule organisation [18]. Prior to fusion into multi-nucleated myotubes, proteins that seed and anchor microtubules, such as γ-tubulin, pericentrin and ninein, relocalise from the pericentriolar matrix to nuclear and Golgi membranes, centriole pairs split due to the cleavage of protein tethers and centrosomes are reduced in size and disassembled (Figure 2). The reduction in centrosomes would be expected to trigger cilia disassembly, but whether the two processes are linked and how they contribute to myogenesis are not well understood.

One possibility is that ciliogenesis may help sequester a proportion of myoblasts in a quiescent state but with a retained proliferative capacity, while the reduction in centrosomes is associated with a transition to a terminally differentiated state for myotube/myofiber formation. In support of this, quiescent satellite cells in vivo or in ex vivo myofiber explants retain a primary cilium, and this is disassembled as satellite cells re-enter the cell cycle to divide and regenerate muscle in response to experimental cardiotoxin-induced myofiber necrosis [10]. Cilia are briefly reassembled in differentiating MyoD^+ve^ myoblasts but are absent in myofibers (Figure 1C), consistent with their disassembly prior to or shortly following myoblast fusion [10]. The appearance of the primary cilium in myoblast subsets during specific stages of myogenesis indicates multiple potential functions of cilia. In the following sections, we explore in more detail our current understanding of cilia functions in skeletal muscle.

## 3. Cilia and Cell Cycle Regulation in Myoblasts

Ciliogenesis is intimately coupled with the cell cycle. The primary cilium is enriched with G-protein-coupled receptors, receptor tyrosine kinases, chemo- or mechanosensing receptors and downstream signal transduction proteins that relay external cues to cell cycle regulators to ultimately determine cell proliferation, self-renewal and cell fate [19]. Moreover, ciliary trafficking, mediated by intraflagellar transport (IFT) protein complexes, has reported functions in cell cycle regulation [20,21]. Thus, abnormalities in ciliogenesis or cilia-mediated signal transduction often lead to cell cycle dysfunctions that are associated with ciliopathies and tumourigenesis [22,23]. However, the molecular relationship between cilia and the cell cycle is complex and may be contextual, with cell type-specific differences.

Typically, in dividing vertebrate cells, cilia are assembled during G_0_/G_1_ and disassembled prior to mitotic entry [23]. Within stem and progenitor populations, the formation of cilia is associated with reversible quiescence, and their disassembly is thought to be tied to cell cycle re-entry and the reinitiation of division for restorative purposes [6]. In most cells, cilia persist through the duration of G1 and are disassembled at G1-S transition through Aurora A kinase (AURKA)-mediated resorption of the ciliary axoneme [24], and live imaging studies of cultured fibroblasts revealed that cilia resorption was preceded by the extracellular release of a ciliary vesicle from the distal tip, a process termed ‘ciliary decapitation’ [25]. It is unclear whether ciliary decapitation occurs widely and whether it frequently precedes cilia resorption in other cell types. However, it is intriguing to consider whether the release of a ciliary vesicle, and subsequent cilia loss, may trigger emergence from a quiescent state of somatic stem cells, such as satellite cells. In addition to cilia resorption, the cell cycle-associated loss of cilia can also be achieved through the rapid ‘shedding’ of the axoneme due to the activity of microtubule-severing enzymes [26]. These processes culminate in the loss of cilia, which is thought to relieve mitogenic repression and allow mitotic progression [27,28]. For example, the depletion of IFT proteins essential for ciliogenesis promotes cell cycle progression to G2 and mitotic phases [21]; studies showing the consequences of the silencing/depletion of various IFT proteins and primary cilium ablation in myogenic and other cells in skeletal muscle are summarised in Table 1.

The dynamic assembly/disassembly of cilia during skeletal myogenesis is consistent with a role in restraining cell proliferation. As mentioned above, cilia are prominently observed on quiescent satellite cells or immediately following cell cycle arrest as myoblasts differentiate [10,15]. The formation of cilia is involved in the spatiotemporal co-ordination of the Hedgehog signalling pathway, which is critically required for skeletal muscle morphogenesis [31]. In mammals, Sonic hedgehog (SHH) is the most well studied of the Hedgehog family of secreted signalling proteins and binds its cognate receptor, Patched1, a cell-surface transmembrane protein that is localised to cilia (Figure 3). In the absence of SHH, Patched1 prevents the accumulation of a G-protein-coupled receptor, Smoothened, within the primary cilium to maintain inactive Hedgehog signalling [32]. SHH binding to Patched1 leads to the accumulation and elevated activity of Smoothened, which in turn activates Gli family transcription factors to regulate myogenesis (Figure 3) [33].

Hedgehog signalling appears to have pleiotropic functions during myogenesis. In satellite cells, the activation of Hedgehog signalling and the accumulation of Gli transcription factors prevent the accumulation of MyoD and subsequent differentiation [34] to maintain stem-like characteristics. In contrast, Hedgehog signalling has also been implicated in promoting muscle cell fate and the terminal differentiation of skeletal muscle [35,36]. During early myoblast differentiation, Hedgehog signalling through Gli transcription promotes sustained expression of myogenic factors required for myogenesis [15]. The inhibition of ciliogenesis, through the depletion of proteins essential for cilia assembly (e.g., IFT88), promotes a pro-mitotic transcriptome in skeletal myoblasts and an increase in proliferation (Table 1). The genetic or chemical inhibition of cilia formation in myoblasts leads to inhibition of the Hedgehog pathway and the marked reduction in Myog, MyoD and Myf5 [15]. In the absence of cilia, the inability to sustain the expression of myogenic factors results in decreased myogenic differentiation and increased myoblast proliferation. Thus, ciliogenesis and cilia-mediated Hedgehog signalling appears to restrain proliferation, maintain quiescence in satellite cells and promotes the terminal differentiation of myoblasts committed to myogenesis. It is intriguing to speculate that the loss of cilia may trigger re-initiation of the cell cycle and the emergence of satellite cells from a quiescent state, for example, in response to injury.

## 4. Cilia Maintenance and Satellite Cell Fate

Muscle stem and progenitor cells can divide symmetrically to generate two daughter cells of identical cell fate or asymmetrically to produce two daughter cells of distinct cell fate. Some studies suggest that primary cilia and associated Notch and Hedgehog signalling may be involved in determining cell fate during skeletal myogenesis [10,37,38,39].

As mentioned above, satellite cells that maintain the regenerative capacity of skeletal muscle are ciliated. Primary cilia were predominantly detected in Pax7^+ve^ satellite cells but not in activated (Myf5^+ve^) and differentiating (Myogenin^+ve^) cells in normal adult skeletal muscle (Figure 1C). The chemical inhibition of primary cilium formation in isolated myofibers led to reduced numbers of Pax7^+ve^ cells and increased numbers of Myogenin^+ve^ cells, which suggests that cilia are required to maintain muscle stem cells [10]. Moreover, primary cilia were observed on proliferating transit-amplifying myoblasts undergoing symmetric cell division. However, during asymmetric division of satellite cells, the primary cilia are also distributed asymmetrically and associated with the self-renewed Pax7^+ve^ daughter cells [10]. Moreover, a study has shown that the ciliary membrane attached to the mother centriole was endocytosed at the onset of mitosis, localised to one centrosome during mitosis and was asymmetrically inherited by one daughter cell. This daughter cell was able to reassemble a functional primary cilium and retained stem cell character compared to the non-inheriting daughter cell [40]. Thus, primary cilia appear to be asymmetrically inherited by a pool of undifferentiated satellite cells and may contribute to their capacity to self-renew.

Ciliary Hedgehog signalling may act as a gatekeeper in determining cell fate [41,42]. Indeed, the transcription factor Gli3 is processed into its repressor form at the primary cilium (Figure 3), and this appears to keep satellite cells in the quiescent state. Gli3-depleted satellite cells were rapidly activated, entered the cell cycle and underwent symmetrical cell division in the absence of injury or Hedgehog stimulation [43]. Moreover, ciliary Hedgehog signalling has been shown to activate the Gli2 transcription factor, which, associated with the *MyoD* gene elements, induced *MyoD* expression and, therefore, initiated early myogenesis [44]. Furthermore, it is reported that satellite cells in the muscles of old mice showed reduced ciliation, and, upon acute muscle injury, these cells showed reduced regeneration capacity via dampened ciliary Hedgehog signalling response [45]. Thus, ciliary Hedgehog signalling is essential to maintain a balance of quiescent and activated stem cells for appropriate skeletal myogenesis. It is noted that there is controversy disputing the status of the myogenic capacity of satellite cells in very old muscles, with many in vivo studies reporting delayed but excellent intrinsic myogenesis and new muscle formation in old mice [46], and similarly good myogenic capacity was demonstrated for very old human muscles [47].

In addition to satellite cells, other populations of cells in the interstitial connective tissue of skeletal muscles are also ciliated (Figure 4, Table 1). These include mesenchymal stem cells and fibro–adipogenic progenitors (FAPs) that are considered to be very similar or identical to fibroblasts [48] and can differentiate into myofibroblasts or adipocytes [49,50,51]. In addition, cilia play important roles in smooth muscle cells and endothelium in the vasculature [52,53], and cilia on the Schwann cells of nerves are involved in myelination [54]. In contrast, skeletal muscle-resident macrophages, which are functionally involved in regulating muscle growth and regeneration, are not thought to assemble primary cilia [55]. The production of many extracellular matrix (ECM) molecules by fibroblasts/FAPs and myofibroblasts are important for muscle regeneration but can also result in excessive collagen and fibrosis, and fibrosis combined with expanded adipocyte numbers can replace the loss of functional muscle contributing to chronic disease states, such as Duchenne muscular dystrophy (DMD) [56]. Surprisingly, the genetic deletion of IFT88 in FAPs, inhibiting ciliogenesis, resulted in restricted adipogenesis in response to acute injury or in DMD mice, and this augmented myofiber formation [11]. Cilia loss was accompanied by the derepression of Hedgehog target genes to prevent the differentiation of FAPs into adipocytes (Table 1). Thus, the absence of cilia in FAPs attenuated adipogenesis and, together with the remodelling of the ECM (discussed in more detail below), contributed to improved myogenesis in response to injury.

It is useful to also consider the possible contribution of cilia in vasculature and nerves to the loss of function in muscular dystrophies. For example, dystrophic nerves of older rodent models of DMD have increased levels of neuronal proteins suggesting progressive neurodegeneration, evident by 9 months of age [57], and there is increasing interest in the role of Schwann cells in the maintenance of axons, especially via the release of extracellular vesicles (reviewed in [58]): in this context, it is of interest that cilia can produce distinct extracellular vesicles [53]. Taken together, primary cilia in muscle progenitors and non-muscle cells are important determinants of cell fate, function and tissue interactions, and these all contribute to skeletal muscle homeostasis.

## 5. Consequences of Ciliary Dysfunction in Satellite Cells

Consistent with the notion that cilia regulate the balance of myoblast proliferation and differentiation, ciliary and Hedgehog signalling abnormalities have been implicated in rhabdomyosarcoma (RMS), a common paediatric tumour that histologically resembles embryonic skeletal muscle and is thought to arise from a defect in myoblast differentiation and an arrested state of skeletal muscle development [59,60]. The assembly of cilia was shown to be suboptimal or was absent in RMS myoblast lines, and this correlated with a defect in Hedgehog signalling and poor differentiation responses [15]. Interestingly, a subset of RMS cell lines that exhibited cilia, albeit with delayed assembly, was reported to subsequently maintain their cilia that persisted into the late stages of differentiation [15]. As mentioned previously, cilia are typically disassembled in normal skeletal myoblasts during late differentiation and during myotube formation. Hedgehog signalling and Gli expression were also significantly upregulated in these RMS cells but were paradoxically associated with poor myoblast differentiation and failed myogenesis [15]. The disassembly of cilia during late myoblast differentiation is likely a consequence of centrosome reduction and the switch to non-centrosomal microtubule organisation, which occurs during this stage of myogenesis. This is further evidence that the timely formation and resorption of cilia may be critical for the normal progression of myogenesis, although this requires further investigation. In addition, the mechanisms that underlie the defects in cilia assembly/disassembly in RMS cells and the extent that cilia defects contribute to RMS progression remain to be fully defined.

To date, the specific description of the pathological consequences of satellite cell cilia dysfunction has been limited largely to RMS. The concept of some primary intrinsic dysfunction of satellite cells has been proposed for various complex neuromuscular diseases (and during normal ageing), without consideration of cilia, although it seems likely that possible altered myogenesis may instead more widely reflect interactions with an adverse ECM (as discussed below). Nevertheless, the extent that disrupted cilia regulation of satellite cell fate might contribute to common myopathy states more broadly warrants investigation. Moreover, in the context of cilia on satellite cells/myoblasts and neuromuscular diseases, it is relevant to consider the wider impact that some drugs might have on cilial health, especially microtubule-targeting drugs used in chemotherapies.

## 6. Cilia and Myoblast Fusion

The fusion of myoblasts to generate multi-nucleated myotubes that mature into myofibers is critically required for normal muscle development and regeneration [61,62,63]. The process of fusion between myoblasts and myotubes and with existing myofibers is a complex, multifaceted process that involves regulation by numerous signalling proteins, cytoskeletal rearrangements, cellular migration, recognition and plasma membrane mixing to facilitate fusion [14,61]. Importantly, the expression of highly conserved, muscle-specific fusogens, Myomaker and Minion–Myomerger, are sufficient for myoblast fusion, as their ectopic expression can promote the fusion of non-muscle cells, such as fibroblasts [64,65]. These molecular processes are thoroughly explored elsewhere [63,66]. Here, we focus on whether cilia in myoblasts are involved in the fusogenic event during myogenesis.

As mentioned previously, cilia are present on fate-committed myoblasts but are disassembled during myoblast fusion. This occurs in concert with the reduction in centrosomes and the reorganisation of the microtubule and actin cytoskeleton necessary for the formation of multi-nucleated myotubes and sarcomere assembly. It is proposed that γ-actin plays a key early role in initiating sarcomere assembly of sarcomeric proteins during differentiation [67]. Interestingly, there is a well-established antagonistic relationship between the global polymerisation of actin and the capacity for ciliogenesis. Pharmacological inhibition of actin polymerisation with cytochalasin D promotes cilia formation and can rescue a deficiency in ciliogenesis caused by hypomorphic mutations to IFT88 [68]. Conversely, ciliogenesis is prevented with positive regulators of actin polymerisation [68]. This indicates an important role for actin dynamics in the transport of proteins to the ciliary compartment for axonemal growth. Thus, the loss of cilia is consistent with the increased actin polymerisation associated with the formation of muscle stress fibres in multinucleated myofibers as precursors for sarcomere assembly [69]. However, whether cilia are involved in the fusogenic event is unclear and difficult to delineate from closely associated functions in cell cycle regulation and differentiation. With that caveat in mind, however, the genetic ablation of cilia in skeletal muscle, with IFT88 or CEP290 knockout, significantly reduces the percentage of myoblasts undergoing fusion [15]. In addition, a detailed electron microscopy study revealed the presence of cilia at the sites of fusion between murine myoblasts [14]. Cilia are well recognised sensory appendages that respond to a variety of extracellular stimuli and are important determinants of cell polarity. Thus, one could suggest that cilia present on mono-nucleated myoblasts may participate in cellular recognition and the regulation of fusogenic proteins to facilitate fusion. The absence of cilia on multinucleated myofibers may also promote secondary fusion, as ciliated mono-nucleated myoblasts may fuse preferentially with established myofibers.

The disassembly of cilia during myoblast fusion suggests that ciliary functions may not be required for this stage of myogenesis. However, our close inspection of differentiating cultured myoblasts indicates that remnants of the ciliary membrane, as marked by Arl13b, attached to centrioles are clearly visible within multi-nucleated myotubes (Figure 5). Moreover, we observed multiple internalised ciliary remnants within multinucleated myotubes (Figure 5). This indicates that ciliary structures are not completely disassembled and are retained for some time in newly formed myofibers. This is analogous to the retention and internalisation of ciliary membrane remnants during cell division in cultured epithelial cells and radial glia progenitor cells in the developing forebrain [40]. As cilia remnants are specifically inherited by a single daughter, this has the potential for asymmetric signalling and fate consequences between the progeny of cellular mitoses [70]. The functions of remnant cilia are generally underappreciated, and it remains unknown if these structures continue to signal following internalisation. One could speculate that the internalised remnant of primary cilia following myoblast fusion may contribute to the normal development and maturation of newly formed myofibers.

## 7. Cilia Interactions with the Extracellular Matrix (ECM) in Skeletal Muscle

In combination with intrinsic factors, the extrinsic ECM environment in skeletal muscle is an important determinant of adequate myogenesis for development and regeneration [71]. Skeletal muscle myofibers and satellite cells are closely associated with the ECM comprised of the basement membrane (or basal lamina) and interstitial connective tissue that contains blood vessels; nerves and other non-muscle cells, such as fibroblasts; immune cells, such as macrophages; and adipocytes (Figure 4) [71]. The ECM contributes to the biomechanical support and force transduction in skeletal muscle, provides access to blood supply and oxygen and serves as a reservoir for growth factors and morphogens. Along with proteoglycans, integrins and other ECM molecules, which are detected by myofibers and their progenitors, the ECM strongly influences the differentiation, fusion and maturation of myofibers and their responses to inflammation and muscle damage [71]. Given the formation of cilia on satellite cells as they enter quiescence and on differentiating myoblasts, the question arises as to the extent that myogenic cilia interactions with the ECM might influence myogenesis. In the context of other tissues, there is abundant evidence that cilia mediate cellular responses to the ECM and, vice versa, the remodelling of the ECM by ciliated cells [72,73].

The excessive fibrosis and pathological accumulation of ECM proteins observed in ‘ciliopathies’, such as renal cystic and Bardet-Biedl syndromes, suggests a proper cilia–ECM dialog is important for normal cellular functions and organ development. The ECM influences cilia length and ECM components, such as laminin, and collagen has been shown to promote cilia growth [74,75]. There is increasing evidence that cilia signalling influences cellular adhesions and communication with the basement membrane. In addition to receptors that respond to various growth factors and morphogens in the extracellular space, the ciliary membrane, in chondrocytes and osteoblasts, for example, are also comprised of integrin receptors and chondroitin sulphate proteoglycans that respond to ECM proteins for mechanotransduction and proper bone development [76]. Cilia in osteocytes contact collagen fibres in tendons and epithelial cells in hair follicles, and developing mammary glands are oriented towards the ECM and are involved in normal development [77,78]. Ciliated satellite cells in skeletal muscle may similarly ‘scan’ the ECM, and this may influence cellular decisions to proliferate, differentiate or return to quiescence. Collagen plays a crucial role in regulating satellite cell self-renewal [79], and the loss of fibronectin and integrin signalling impairs muscle regeneration [80]. Likewise, other ECM components, such as TGF-β, glycosaminoglycans and hyaluronan, have been shown to inhibit myoblast differentiation and subsequent fusion and myotube formation [71,81]. The ECM in aged skeletal muscle also alters satellite cell differentiation in response to injury, triggering their conversion to fibroblasts to exaggerate fibrosis [82]. Thus, it is likely that the mechano-, chemo-sensory functions of cilia would contribute to ECM influences on satellite cell behaviour and myogenesis, although this has yet to be specifically tested. It is well established that satellite cells respond to ECM remodelling during muscle adaptations to ageing and to damage and inflammation during muscular dystrophies [71].

In addition to sensing the extracellular environment, ciliary signalling in satellite cells and differentiated myoblasts may also contribute to remodelling of the ECM. Hedgehog signalling in chondrocytes brings about changes in ECM composition associated with bone development [83]. Similarly, proteoglycan and collagen expression in cartilage during growth plate development is altered in the absence of cilia from the depletion of IFT88 or KIF3A [84,85]. Changes in matrix composition in the absence of cilia may arise in large part from indirect mechanisms [72]. However, ECM genes and TGF-β signalling targets are known direct downstream transcriptional targets of cilia signalling [86,87]. Moreover, ciliary signalling is linked to the enhanced expression and secretion of proteases, such as matrix metalloproteinases (MMPs) and disintegrins and metalloproteinases (ADAMs), as well as tissue inhibitors of metalloproteinases (TIMPs), to regulate matrix degradation [72]. Likewise, satellite cells have been shown to be an autocrine source of MMP13 [88], and subsequent remodelling of the ECM by MMP13 was required to support myoblast migration for muscle growth and repair [88]. At the time of writing this review, the role of satellite cell ciliary signalling in ECM regulation has not been specifically investigated.

Myogenic stem cells may regulate ECM composition in a cell-autonomous or non-cell autonomous fashion. It is well established that FAPs/fibroblasts and myofibroblasts represent the main protagonists in fibrosis of skeletal muscle [71,81] and other tissues [89]. While issues of mechanosensing and mechanotransduction are widely discussed in this context [90], the myofibroblast cilium plays a major role in the signalling associated with the process of fibrosis, especially since several components of the TGF-β1 pathway are located within the cilium [30]. The ECM modifying activity of fibrogenic cells is also influenced by satellite cells. Satellite cells secrete exosomes that transmit microRNAs to fibroblasts to modify collagen synthesis, and this prevents excessive ECM deposition for optimal muscle repair [91]. Thus, satellite cells may secrete matrix proteins but also modify their ECM environment indirectly through interstitial fibrogenic cells. In addition, in response to skeletal muscle damage, cross-talk between fibroblasts and inflammatory cells within the endomysium significantly influences the deposition of fibronectin, collagen and other ECM proteins [92,93]. It seems likely that ciliary signalling in skeletal muscle FAPs/fibroblasts may regulate the ECM to modify myogenic responses (Table 1). Whilst cilia motility defects have long been recognised in the pathology of cystic fibrosis [94], a role for primary cilium in fibrosis in other tissues is also evident [95]. The numbers of primary cilia and Hedgehog pathway signalling are increased to stimulate fibroblast production of ECM during pulmonary fibrosis [96]. Following myocardial injury, cilia on cardiac fibroblasts are similarly required for TGF-β-induced fibrosis [97]. In skeletal muscle, ciliary Hedgehog signalling on FAPs upregulates the expression of TIMP13, an inhibitor of MMP14, to remodel the ECM, and this restricts adipogenesis during muscle repair in a cell non-autonomous manner [11]. Following injury in disease contexts, the repression of ciliary Hedgehog-mediated upregulation of TIMP3 results in excessive adipogenesis and reduced myogenic repair [11]. These studies suggest that feedback mechanisms between cilia and the ECM may act in a cell autonomously or non-autonomously manner in satellite cells and FAPs/fibroblasts resident in connective tissue as important mediators of muscle growth, repair, homeostasis and ageing. They also contribute myoblasts to muscle regeneration in response to myonecrosis resulting from either intrinsic muscle degeneration (as in DMD) or experimental or accidental injury.

## 8. Conclusions and Future Directions

In skeletal muscle tissue, primary cilia are clearly present on skeletal muscle progenitors, quiescent and activated satellite cells and myoblasts and differentiating myoblasts (but not on proliferating myoblasts), and on fibroblasts/FAPs and other interstitial cells, as well as cells associated with blood vessels and nerves. The important functions of cilia in cell cycle regulation and cell fate specification are now evident, and emerging evidence hints at undescribed functions in myoblast fusion and the regulation of ECM organisation in the skeletal muscle microenvironment surrounding myofibers and satellite cells. A complete picture of cilia function in skeletal muscle health and disease is yet to fully emerge. To date, no studies have yet been attempted to specifically ablate cilia in satellite cells, and the feedback interactions between cilia on various cell populations in skeletal muscle tissue require more extensive interrogation. Finally, the intrinsic contribution of cilia in satellite cells/myoblasts or extrinsic roles in non-muscle cells within the context or normal ageing muscles and sarcopenia and neuromuscular diseases, such as muscular dystrophies, remains undefined. It is likely that more consideration of the contribution of cilia in these situations and a better delineation of their roles related to modulation and interactions between myogenesis and the ECM may lead to an improved understanding of disease mechanisms or novel treatment strategies.

## Figures and Tables

**Figure 1 ijms-22-09605-f001:**
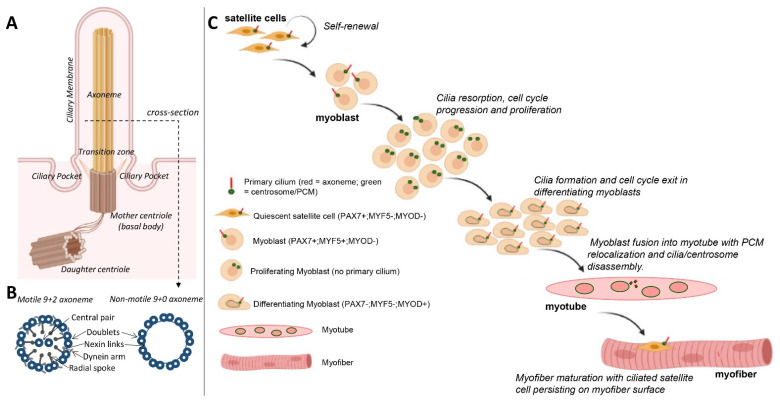
Primary cilia assembly and disassembly during myogenesis. (**A**) The mother centriole with distal appendages forms the basal body of primary cilia. The distal appendages delineate a transition zone and a ciliary compartment distinct from the cytoplasm. Axonemes are comprised of bundles of microtubules that extend upwards from the basal body and are encased in a ciliary membrane. (**B**) Cross-sectional view of axoneme. Motile cilia possess a 9 + 2 radial configuration of axonemal microtubules. Outer doublet microtubules are connected by nexin links and interact with central single microtubules via radial spoke complexes. Non-motile primary cilia in muscle are organised in a 9 + 0 configuration. (**C**) Cilia formation and resorption during myogenesis. Satellite cells expressing PAX7 (PAX7^+ve^) are ciliated, and these are resorbed as satellite cells specify proliferative myoblasts that express Myf5. Cilia are briefly reassembled in myoblasts as they downregulate PAX7, upregulate MYOD (MYOD^+ve^) for differentiation and exit the cell cycle. During late stage differentiation and fusion into myotubes, cilia are disassembled, and centrosomes undergo reduction. Cilia are largely absent in mature myofibers but found on quiescent satellite cells located between the myofiber and basal lamina.

**Figure 2 ijms-22-09605-f002:**
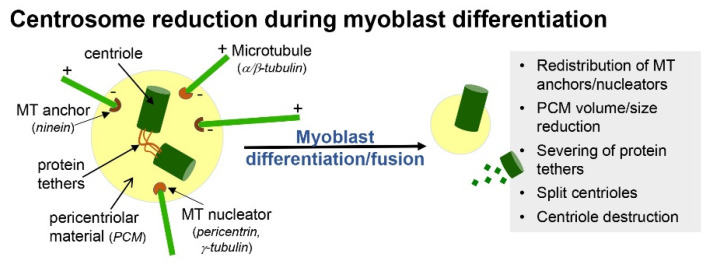
Centrosome reduction during myoblast differentiation. The inactivation of and reduction in centrosomes, which serve as basal bodies for cilia, accompany the terminal differentiation of myoblasts and fusion into myotubes. Centrosome volumes are reduced and pericentriolar material, comprising microtubule anchoring/nucleating complexes, are redistributed to other cellular compartments. In addition, proteins that tether centriole pairs are severed, and centriole barrels are disassembled.

**Figure 3 ijms-22-09605-f003:**
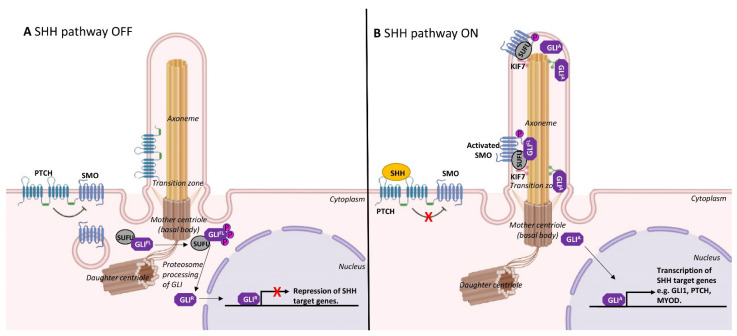
Hedgehog signalling on primary cilia. (**A**) In the absence of Sonic hedgehog (SHH), Patched1 (PTCH1) inhibits Smoothened (SMO) and prevents SMO localisation to the primary cilium. Suppressor of Fused (SUFU) prevents activation of full-length GLI (GLI-FL) transcription factor, leading to proteosome processing of GLI-FL to its repressor form (GLI-R). GLI-R translocates into the nucleus to repress the expression of SHH target genes. (**B**) In the presence of SHH, PTCH1 releases its inhibition to SMO. SMO is then activated and localises to the primary cilium along with KIF7, SUFU and full-length GLI (GLI-FL). Activated SMO allows SUFU to release GLI-FL at the tip of the cilium to allow activation of GLI (GLI-A). GLI-A then translocates into the nucleus and transcribes SHH target genes, such as GLI1, PTCH1 and Myogenic differentiation 1 (MYOD).

**Figure 4 ijms-22-09605-f004:**
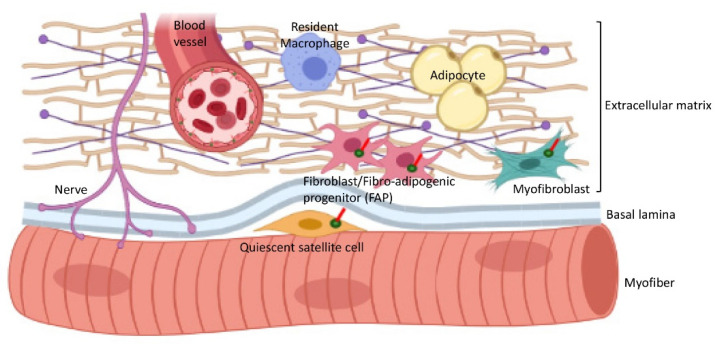
Diagram of skeletal muscle indicating cilia of a satellite cell on a myofiber surface and cilia of various cells in interstitial connective tissue. In the adult skeletal muscle, the quiescent satellite cell is on the myofiber and under the basal lamina. The myofiber is surrounded by the extracellular matrix, which also contains blood vessels, nerves, macrophages, fibroblast/fibro–adipogenic progenitor (FAP), adipocytes and myofibroblasts.

**Figure 5 ijms-22-09605-f005:**
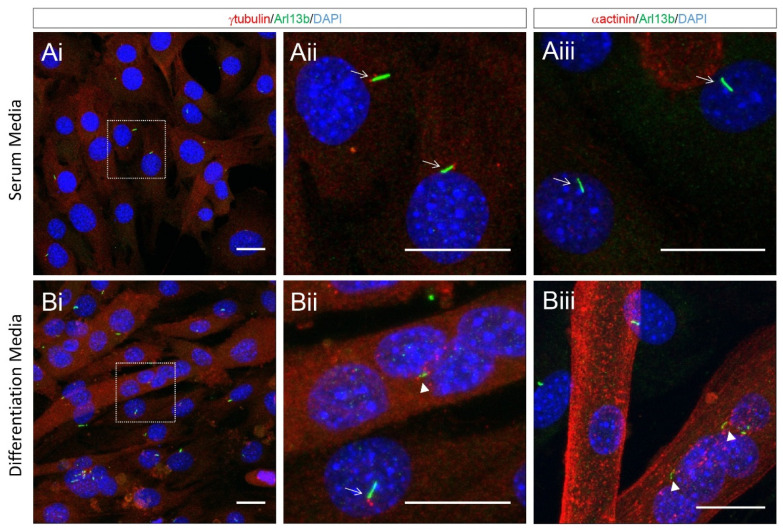
Cilia resorption and cilia remnants in newly formed myotubes. Tissue cultures of C2C12 mouse skeletal myoblasts during (**A**) proliferation (cultured in 10% fetal bovine serum) and (**B**) after 3 days in differentiation medium (2% horse serum) to induce fusion to form myofibers. Cells are immunostained to show cilia (green Arl13b) and basal body (red γ-tubulin) in (**i**) and (**ii**), respectively, or in (**iii**) only for cilia (green Arl13b) and counterstained with actinin (red) to show differentiating myoblasts and myotube sarcoplasm. Cells observed by fluorescent microscopy. Primary cilia are evident in (**A**) undifferentiated (actinin^−ve^), mono-nucleated myoblasts. Cilia are disassembled in (**B**) as myoblasts differentiate (actinin^+ve^) and fuse into multinucleated myotubes. Line arrows indicate cilia in myoblasts. Arrowheads indicate cilia remnants in myotubes. Scale bars = 20 μm.

**Table 1 ijms-22-09605-t001:** Consequences of primary cilium ablation in skeletal muscle cell populations.

Cell Type	Targeting	Cellular Phenotype and Signalling Consequences	Reference
C2C12 and primary murine myoblasts	miRNA silencing of CEP290, IFT80 or IFT88 Ciliobrevin D	Increased proliferation. Reduced differentiation and MRF (MyoD, Myf5) expression. Reduced myoblast fusion and myogenesis. Inhibited Hedgehog signalling.	[15]
C2C12 myoblasts	siRNA silencing of IFT88	Altered quiescence. Reduced self-renewal potential. Enhanced progression to G2/M. Enhanced growth factor signalling to mTOR.	[29]
Pax7^+ve^ satellite cells within isolated myofibers	Nocodazole or Taxol. Forchlorfenuron (septin inhibitor)	Reduced self-renewal. Increased myogenin expression.	[10]
Fibro–adipogenic progenitors (in vivo)	*Pdgfra*-CreERT deletion of IFT88.	Decreased differentiation to adipocytes and reduced adipogenesis. Increased myofiber size in *Dmd^mdx^* and following injury. Derepressed Hh target genes (Gli1 and Ptch1). Enhanced TIMP3 expression.	[5]
Adipose progenitors (in vitro)	siRNA silencing of Kif3A. Ciliobrevin D	Reduced differentiation to myofibroblasts. Inhibited TGFβ signalling.	[30]
Myofibroblasts (in vitro)	Ciliobrevin D	Decrease in myofibroblast phenotype.	[30]

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
