# Peer review of "Cilia, Centrosomes and Skeletal Muscle"

_ijms, 2021, doi:10.3390/ijms22179605_

Round 1
Reviewer 1 Report
In this review, Ng et al. have provided a comprehensive overview for the role of primary cilia in cell cycle regulation, cell signaling and specifically addressing the formation and resorption of cilia during skeletal myogenesis. The authors also summarized the up-to-date experimental evidence for the role of primary cilia in satellite cell fate determination and myoblast fusion, the interaction of cilia with extracellular matrix in skeletal muscle, as well as the consequences of ciliary dysfunction in satellite cells. Overall this review article is very well written and well organized. Figure illustrations are informative, and intuitively well received. I have only several minor comments.
- Line 67, line 69 and throughout the text, it is not clear about the meaning of “+ve” in PAX7+ve and MYOD+ve.
- Line 177 and Figure 2, the abbreviation of sonic hedgehog was first presented as “Shh”, but in Figure 2 and later paragraphs it was presented as “SHH”. It would be better to unify the abbreviation of sonic hedgehog throughout the text.
- Line 320, “there is a well establish antagonistic”, “establish” should be changed to “established”.
- Line 359, “gtubulin”, g is in red, should be changed to black.
- Line 363, “scale bars = 20 uM”, should be changed to “scale bars = 20 um”.
Author Response
We would like to thank Reviewer 1 for a favourable assessment of our manuscript and for pointing out several minor text errors.
We have addresses all the issues raised in our revised manuscript with our changes indicated in the revision with yellow highlight.
We have also responded to each reviewer points below:
- Line 67, line 69 and throughout the text, it is not clear about the meaning of “+ve” in PAX7+ve and MYOD+ve. We have expanded slightly our text to explain that this refers to cells expressing PAX7 or MYOD in the first instance where these notations appear.
- Line 177 and Figure 2, the abbreviation of sonic hedgehog was first presented as “Shh”, but in Figure 2 and later paragraphs it was presented as “SHH”. It would be better to unify the abbreviation of sonic hedgehog throughout the text. Done
- Line 320, “there is a well establish antagonistic”, “establish” should be changed to “established”. Done
- Line 359, “gtubulin”, g is in red, should be changed to black. Done
- Line 363, “scale bars = 20 uM”, should be changed to “scale bars = 20 um” Done
Reviewer 2 Report
General
So far, it is poorly understood what mechanisms recruit, or fail to recruit satellite cells for muscle regeneration and disease and ageing states. I congratulate the authors to have put together this thought-provoking and conceptionally strong review, with a clear focus on cilia assembly, integrity, dis-assembly, and the relationship to satellite cells and muscle regeneration. The concept that cilial integrity regulates the balance of myoblast proliferation and differentiation has important implications for muscle health.
Specific comments
Section 4, title Cilia can control cell fate, and later line 216 – may be involved in controlling cell fate. I suggest to revise the title to be more in line with line 216, such as to a title Cilia maintenance and cell fate, or cilia may control cell fate (i.e. a downtuned title statement)
Section 5, consequences on cilial dysfunction.
This section is fully focused on the rare rhabdomyosarcomas.I wonder if instead the authors should focus, or at least add considerations, on cilial dysfunction in other, more common myopathy states (speculative or supported by references). And then perhaps rename the section to “potential consequences of cilial dysfunction in muscle health”.
Potential fields that could be discussed are for example drugs used in chemotherapies that disrupt microtubuli (and do cause neuromuscular side effects), the concepts to treat muscular dystrophies, such as with grafts, ES cells (has their appropriate cilial state been considered?), perhaps the need when treating DMD with the novel emerging drugs, that cilial health has also to bee considered, and not just fixing a molecular pathway.
These are just ideas, but perhaps the authors agree that a more general medical myology relevance outlook discussion might fit better here instead of a specialized RMS-only discussion that is not of great interest to most readers. The revised and expanded discussion could perhaps be moved to section 8, as an expanded outlook section on cilia integrity and function in muscle health and diseases.
Author Response
We would like to thank reviewer 2 for their positive comments and favourable assessment of our manuscript. Our response to the specific suggestions are as follows:
Comment: Section 4, title Cilia can control cell fate, and later line 216 – may be involved in controlling cell fate. I suggest to revise the title to be more in line with line 216, such as to a title Cilia maintenance and cell fate, or cilia may control cell fate.
Our Response: This is an acceptable suggestion by the reviewer. We have edited the section title as suggested.
Comment: Section 5, consequences on cilial dysfunction. This section is fully focused on the rare rhabdomyosarcomas.I wonder if instead the authors should focus, or at least add considerations, on cilial dysfunction in other, more common myopathy states...
Our Response: We appreciate the reviewer's perspective that this section read with perhaps too narrow a focus. However our specific focus on RMS was due to limited published studies that have looked specifically at satellite cilia in this context and related specifically to intrinsic proliferation/differentation regulation discussed in the preceding section (Section 4). Therefore, we feel it is appropriate to leave this section where it is. In subsequent sections we discuss cilia interactions with ECM in relation to muscle dystrophies. To address Reviewer comments, we have instead expanded on this section and included a new paragraph echoing reviewer sentiments that cilia health may have broader impacts and should be considered in muscle health more broadly. Our changes highlighted in revised manuscript in yellow highlight. A tantalizing area for future investigation for certain. We trust this is acceptable for the reviewer.